# Frail Silk: Is the Hughes-Stovin Syndrome a Behçet Syndrome Subtype with Aneurysm-Involved Gene Variants?

**DOI:** 10.3390/ijms24043160

**Published:** 2023-02-05

**Authors:** Simona Manole, Raluca Rancea, Romana Vulturar, Siao-Pin Simon, Adrian Molnar, Laura Damian

**Affiliations:** 1Department of Radiology, “Niculae Stăncioiu” Heart Institute, 19-21 Calea Moților Street, 400001 Cluj-Napoca, Romania; 2Department of Radiology, “Iuliu Hatieganu” University of Medicine and Pharmacy, 400012 Cluj-Napoca, Romania; 3Cardiology Department, Heart Institute “Niculae Stăncioiu”, 19-21 Calea Moților Street, 400001 Cluj-Napoca, Romania; 4Department of Molecular Sciences, “Iuliu Hatieganu” University of Medicine and Pharmacy 6, Pasteur, 400349 Cluj-Napoca, Romania; 5Cognitive Neuroscience Laboratory, University Babes-Bolyai, 30, Fântânele Street, 400294 Cluj-Napoca, Romania; 6Department of Rheumatology, Emergency Clinical County Hospital Cluj, Centre for Rare Autoimmune and Autoinflammatory Diseases (ERN-ReCONNET), 2-4 Clinicilor Street, 400347 Cluj-Napoca, Romania; 7Discipline of Rheumatology, “Iuliu Hațieganu” University of Medicine and Pharmacy, 400347 Cluj-Napoca, Romania; 8Department of Cardiovascular Surgery, Heart Institute “Niculae Stăncioiu”, 19-21 Calea Moților Street, 400001 Cluj-Napoca, Romania; 9Department of Cardiovascular and Thoracic Surgery, “Iuliu Hatieganu” University of Medicine and Pharmacy, 8 Victor Babes Street, 400012 Cluj-Napoca, Romania; 10CMI Reumatologie Dr. Damian, 6-8 Petru Maior Street, 400002 Cluj-Napoca, Romania

**Keywords:** Hughes-Stovin syndrome, Behçet syndrome, vascular cluster, pulmonary artery aneurysm, *MYLK*, myosin light chain kinase, thoracic aortic aneurysms and dissections, personalized medicine

## Abstract

Hughes-Stovin syndrome is a rare disease characterized by thrombophlebitis and multiple pulmonary and/or bronchial aneurysms. The etiology and pathogenesis of HSS are incompletely known. The current consensus is that vasculitis underlies the pathogenic process, and pulmonary thrombosis follows arterial wall inflammation. As such, Hughes-Stovin syndrome may belong to the vascular cluster with lung involvement of Behçet syndrome, although oral aphtae, arthritis, and uveitis are rarely found. Behçet syndrome is a multifactorial polygenic disease with genetic, epigenetic, environmental, and mostly immunological contributors. The different Behçet syndrome phenotypes are presumably based upon different genetic determinants involving more than one pathogenic pathway. Hughes-Stovin syndrome may have common pathways with fibromuscular dysplasias and other diseases evolving with vascular aneurysms. We describe a Hughes-Stovin syndrome case fulfilling the Behçet syndrome criteria. A *MYLK* variant of unknown significance was detected, along with other heterozygous mutations in genes that may impact angiogenesis pathways. We discuss the possible involvement of these genetic findings, as well as other potential common determinants of Behçet/Hughes-Stovin syndrome and aneurysms in vascular Behçet syndrome. Recent advances in diagnostic techniques, including genetic testing, could help diagnose a specific Behçet syndrome subtype and other associated conditions to personalize the disease management.

## 1. Introduction

Hughes-Stovin syndrome (HSS) is a rare disease characterized by widespread thrombosis and multiple pulmonary and/or bronchial aneurysms [1,2]. HSS is often considered to be a variant of Behçet syndrome (BS), the “silk road disease” [1,2]. There are only about 90 HSS cases reported in the literature [3]. Generally, pulmonary artery aneurysms (PAA) are uncommon and may be asymptomatic, but may result in rupture or dissection and may have genetic basis [4]. There is a lack of HSS diagnostic criteria, and generally, a vascular occlusive disease (venous and/or arterial) with a normal coagulation profile and PAA with thrombosis are found [3,5]. The aneurysm–thrombosis combination with a negative infectious screening suggests HSS or BS [6]. The diagnosis relies on computed tomography (CT) pulmonary angiography showing PAA with adherent in situ thrombosis, according to the criteria of the HSS International Study Group [5].

The etiology and pathogenesis of HSS are incompletely known. Vasculitis underlies the pathogenic process, while infections and angiodysplasia may also contribute [1]. The condition is classically described to evolve in 3 phases: thrombophlebitis, formation of pulmonary and/or bronchial aneurysms, and aneurysm rupture leading to hemoptysis [1]. Nevertheless, vasculitis is an early finding, and PA thrombosis likely develops secondary to vessel wall inflammation [3,5]. In BS, the PAA are due to the obliterative endarteritis of the vasa vasorum, or they are rather pseudoaneurysms due to vessel wall edema, usually formed after perforation [2,7,8,9]. Hemoptysis, the dreaded complication in HSS, is likely due to the rupture of angiodysplastic bronchial arteries, but also lobal/segmental PA vasculitis [6,10]. Besides lung involvement, vasculitis complicated by aneurysms may involve any vessel [5,8,11]. In HSS, the histology reveals diffuse dilatation, partial occlusion, inflammatory cell infiltration, and destruction of elastic and muscular fibers in the vessel wall [1,12,13,14,15]. BS also has similar histologic characteristics, and vasa vasorum vasculitis leads to elastic fibers destruction and aneurysm formation as well [16,17,18]. The combination of vasculitis involving the arteries and veins, thrombosis, and aneurysms suggests BS [6,17,19]. The term angio-BS or vascular BS defines the disease subset with large vessel involvement predominant in the clinical picture [17,19]. HSS is likely related to the vascular phenotype of BS, and vascular involvement in BS may precede other disease features [7,17]. Isolated pulmonary artery thrombosis also belongs to the vascular cluster of BS with lung involvement [20]. Interestingly, BS features such as oral aphtae, uveitis, and arthritis are rarely seen in HSS, and the patients are most often males [5].

As such, vascular BS and HSS may share pathways with (acquired) vascular dysplasia, in predisposed hosts, in the same manner in which the articular BS clusters may share susceptibility genes and inflammatory pathways to spondylarthritis. However, no such genetic risk factors have been identified in BS to date [21,22]. Therefore the objective of this case report was to find out more about possible inborn connective tissue defects underlying the propensity for PAA in HSS/BS. Here we report the finding of variants of genes involved in angiogenesis in this HSS/BS case.

## 2. Case Presentation

A 35-year male with a history of smoking (10 cigarettes/day for 14 years, or 7 pack-years) was admitted to the local hospital for recurrent hemoptysis. He denied any similar problems among his relatives. His father had suffered from bone cancer and died four years prior. His mother and younger brother were living abroad; they were healthy, apart from his mother’s hypertension.

Two years previously, our patient had been diagnosed by a computed tomography (CT) angiography scan with a low-risk bilateral pulmonary embolism (PE) affecting the segmental and subsegmental arteries. Other five segmental and subsegmental PA in both lungs with adherent thrombi, an inferior vena cava thrombosis from its origin that extended to the left renal vein, and ectasia of the left common iliac vein were seen as well. Thrombophilia testing performed at that time was negative for antiphospholipid syndrome but revealed a heterozygous *PAI-1* variant and a *MTHFR* gene C677T polymorphism, indicating a mildly increased thrombotic risk. The inflammatory markers (erythrocyte sedimentation rate, C reactive protein, and leukocytosis) were also elevated in the absence of any infection. He had been discharged on treatment with rivaroxaban (a factor Xa inhibitor) and low-dose aspirin anti-aggregation.

During the hospitalization for hemoptysis, another CT scan showed no signs of acute PE but described a mass in the apical lower left lobe. A bronchoscopy with biopsy found a vegetant, hemorrhagic mass obstructing the 8th segment’s bronchi. The hemorrhage was stopped with difficulty. The pathology exam revealed a bronchi epithelium intensely infiltrated with polymorphonuclear cells. A week after the bronchoscopy, he repeated the hemoptysis while infected with SARS-CoV-2, for which dexamethasone treatment was initiated. Aspirin was stopped, and he was switched onto a prophylactic dose of enoxaparin (a low-molecular-weight heparin). The CT scan actually raised suspicion of HSS. A positron emission tomography scan done afterward established that the lung mass was not a lung tumor but an aneurysmal dilatation of the inferior left lobar pulmonary artery, with no increased enhancement, based on the normal fluorodeoxyglucose (FDG) uptake. A cardiologic evaluation did not find any intracardiac masses on echocardiography, nor pulmonary hypertension or signs of heart failure. A CT scan was repeated two months after the first hemoptysis episode. The inferior left lobar PAA was stable and decreased in size; there were no signs of alveolar hemorrhage. The rheumatologic evaluation confirmed the diagnosis of HSS; he admitted having a history of oral ulcers since childhood and genital ulcers since the PE episode and tested positive for HLA-B51.

The patient was started on immunosuppression with pulsed cyclophosphamide and methylprednisolone, and, after six months, was switched to azathioprine and oral methylprednisolone with tapering. He had a single episode of mild hemoptysis, rather a haemoptoic sputum, two months after therapy initiation, which was not repeated afterward; the CT scan performed 6 months after showed complete disappearance of the PAA and thrombus, as well as remission of the PA wall thickening. However, the inferior vena cava thrombosis persisted despite immunosuppressant therapy and anticoagulant treatment with low-dose dabigatran (2 × 110 mg/day).

A genetic testing (Illumina NGS, sequence analysis and deletion/duplication testing of connective tissue disorders panel, 92 genes, and inborn errors of immunity and cytopenia panel, 562 genes, respectively, Invitae Corp., San Francisco, CA, USA) identified a heterozygous variant of unknown significance (VUS) in *MYLK*, exon 11, c.1472A>G (p.Asn491Ser), not previously reported in individuals with MYLK-related conditions, not expected to disrupt the MYLK protein function, but able to create or strengthen a splice site according to predictive algorithms developed (ref. UNIPROT, CLINVAR) (Figure 1).

Additionally, heterozygous pathogenic variants were identified in *CR2* and *CFTR* (low penetrance), as well as other heterozygous VUS in *GGCX*, *DNAAF4*, *FANCE,* and *NHP2* genes.

## 3. Discussion

The vascular phenotype of BS, including HSS, has different clinical presentations, presumably based on different genetic determinants [17]. In BS, arterial involvement occurs in 3–5% of patients, and aneurysms interesting pulmonary, visceral, or peripheral arteries represent 60% of the arterial lesions [19,24]. Our HSS case fulfilled the criteria for BS, in whom pulmonary vasculitis underlies the PAA. The initial presentation mimicked a vascularized bronchial tumor. The lack of increased FDG signal enhancement in the PAA walls was likely due to the glucocorticoid therapy given for COVID-19 before the PET-CT scan.

PAA due to underlying vasculitis are the deadliest lesions in BS and are generally associated with peripheral vascular disease [20,25]. Aortic involvement [mainly abdominal] is the most common site of BS arterial involvement, followed by PAA, but other arterial peripheral involvements and intracerebral aneurysms (ICA) are also reported [24].

Generally, the genes rendering patients susceptible to thoracic aortic aneurysms or dissections (TAAD) may increase the risk for other vascular diseases, such as abdominal aortic aneurysms, cerebral, coronary artery aneurysms, and others [26]. TAAD genetic susceptibility is often transmitted autosomal dominant (AD) with decreased penetrance and variable expressivity [23,27,28]. PAA may share TAAD predisposing genes mutations interesting the transforming growth factor beta (TGFβ) signal, extracellular matrix (*FBN1*, *TGFBR1*, *TGFBR2*, *SMAD3*, *TGFB2*, *COL3A1*), and altered components of the contractile apparatus of the smooth muscle cells (SMC): *ACTA2*, *MYH11*, *MYLK,* and *PRKG1* [4,29].

PA dilatation was described in the setting of syndromic TAAD-associated mutations [4,30,31,32,33,34,35,36,37,38]. Our patient had no dysmorphic signs or features to suggest inherited connective tissue disorders but had a *MYLK* VUS.

### 3.1. Could a MYLK Variant Be Involved in the Occurrence of PAA in BD/HSS?

*MYLK* involved in TAAD [39] was not described to date, to our knowledge, in relation to PAA.

The *MYLK* gene (OMIM 600922), located on 3q21, encodes at least 3 proteins (Figure 2) via different unique promoters: non-muscle MLCK 210 (nmMLCK), smooth muscle MLCK 108, and telokin/KRP [28,40,41]. MLCK plays an important role in maintaining SMC contractility and cell survival, but also in cell division, cell migration, and cell–matrix adhesion [42,43,44]. Moreover, *MYLK* regulates tight junctions and microvascular permeability and is involved in fibroblast apoptosis and epithelial wound healing [45]. *MYLK* also regulates actin-myosin interactions through a non-kinase activity [45]. Telokin modulates SMC contraction by inhibiting the myosin RLC (regulatory light chain) phosphatase [46].

Aneurysm formation involves a succession of hemodynamic stress, thrombosis, extracellular matrix (ECM) degradation, inflammation, and structural changes, including endothelial cell (EC) dysfunction and SMC apoptotic and phenotypic modulation [42,47]. Besides the structural role, SMC are involved in vasomotricity due to the contractile proteins, using cross-bridge cycling between actin and myosin, intracellular Ca^2+^ concentration increase, and Ca^2+^ binding to calmodulin to initiate the SMC contraction [27,48,49]. The Ca^2+^-calmodulin complex binds to myosin light chain kinase (MLCK) to activate it, and MLCK phosphorylates the RLC of myosin in turn, which increases the actin-activated myosin II ATPase activity for contraction [27]. The myosin light chain phosphatase dephosphorylates the myosin RLC to induce relaxation [43].

The non-muscle myosin light chain kinase (nmMLCK) is a 210 kDa cytoskeletal protein (Figure 3), central for the regulation of vascular integrity and permeability by regulating actin cytoskeleton rearrangements and contraction, vascular endothelial barrier, angiogenesis, EC apoptosis, and neutrophil transmigration and diapedesis [40,50,51].

The smooth muscle MLCK and nmMLCK share identical c domains, whereas the N terminal domain is unique to nmMLCK and undergoes posttranslational phosphorylation [52]. The Rho kinase may phosphorylate non-muscle myosin in other cell types [43,53]. There are significant differences in MYLK activity in smooth, skeletal, and cardiac muscles [43].

Certain VUS in genes associated with heritable vascular diseases may be low-penetrant “risk variants”, which may result in disease in the presence of other genetic or environmental factors or due to stochastic events [54,55]. *MYLK* haploinsufficiency specifically involves the ascending aorta and not other tissues, with much lower MLCK requirements [43]. *MYLK* mutations associated with TAAD are located in the short form of MLCK (aa 923–1914), the only form expressed in the human aorta [28,29,46,52]. As such, rare variants disrupting amino acids 1 to 922 (like in our case) should not cause aortic aneurysms but may have other vascular consequences [28,29,46,52] (Figure 1).

The clinical phenotype of *MYLK* mutations is not well characterized besides TAAD, as they are not associated with morphological changes, including aortic ectasia [43,56]. Nevertheless, the *MYLK*-related phenotype is expanding. *MYLK* homozygous mutations were described in the megacystic microcolon intestinal hypoperistalsis syndrome [57]. Certain *MYLK* polymorphisms may be associated with severe respiratory inflammatory disorders, such as asthma, acute respiratory distress syndrome, etc. [50]. Also, *MYLK* -associated vascular involvement may result in multiple arterial dissections in phenotypes distinct for the homozigosity or heterozigosity of the *MYLK* variant [27,58]. *MYLK* may also be involved in the occurrence of intracerebral aneurysms (ICA) [59].

In our case, the *MYLK* mutation was located in the Ig-like domain3, involved in the EC cytoskeletal functions [23].

### 3.2. MYLK in Endothelial Inflammation, BS, and Aneurysms

The EC cytoskeleton is involved in vascular barrier integrity and repair [60]. The nmMLCK regulates endothelial and vascular permeability, promoting EC cytoskeleton rearrangements [52,61]. Regulatory mRNAs controlling nmMLCK expression are triggered in response to inflammatory stimuli [50,61]. TNFα increases *MYLK* transcription in lung EC [61], while the transcription factor NRF2 represses it [50]. Epigenetic modification of cytoskeletal dynamics is also important in BS [62]. A ruptured aneurysm involves a vessel wall structure injury or EC apoptotic death, which can be initiated by tumor necrosis factor alpha (TNFα) in BS [42,56].

There are common pathways and mechanisms, some including *MYLK*, in aneurysm formation and BS pathogenesis (Table 1).

*MYLK* is involved in inflammatory responses such as EC apoptosis, vascular permeability, and leukocyte diapedesis [45]. Neutrophils are central in many diseases evolving with inflammation and tissue remodeling, including aneurysms, by releasing neutrophil extracellular traps (NETs) [63]. Neutrophils are key players in BS [64,65]. Adherent neutrophils activate endothelial MLCK, increasing EC contractility and intercellular gaps and thus facilitating neutrophil migration to the inflammatory sites [65,66,67]. Also, *MYLK* triggers neutrophil transmigration by activating integrin β2 in acute lung injury [68].

MLCK is critical in the TNFα-induced EC apoptosis through caspase activation [69,70]. In BS, TNFα results in EC apoptosis and induces the expression of proinflammatory mediators, including metalloproteinases MMP-2 and MMP-9, which are important in ECM destruction and aneurysm formation [42,56,71,72,73,74]. Other factors involved in vascular remodeling, such as mechanical stretching, are intermingled [75,76].

The vascular endothelial growth factor (VEGF), a proangiogenic glycoprotein involved in many cellular processes such as cell migration, proliferation, and angiogenesis, also increases EC permeability [40]. VEGF increases both nmMYLK gene product through the Sp1 transcription factor and nmMLCK enzymatic activity [40]. In BS, the VEGF levels are increased, correlated with the disease activity mostly in vascular BS [77,78]. Nevertheless, VEGF inhibition may result in aneurysms or dissections [79]. The ubiquitin-proteasome system (UPS), involved in SMC inflammation and phenotypic switch, is important in the common pathogenesis of aneurysms [50] and BS [80]. MLCK also intervenes in UPS regulation [81].

From the mitogen-activated protein kinases (MAPK), a family of kinases regulating cell growth, differentiation, and inflammation, ERK (extracellular signal-regulated kinase) 1/2 is increased in BS in EC [81]. ERK activates MMPs during AAA formation [82]. ERK signaling inhibits MLCP in the MLCK/MLCP balance [83].

**Table 1 ijms-24-03160-t001:** MYLK effects and possible pathogenetic involvement in BS/HSS.

*MYLK*/MLCK Function	In BS/HSS
*MYLK* is involved in aneurysm formation [27]	Not previously described in BS
MLCK is critical in the TNFα -induced EC apoptosis (through caspase activation) [69,70]	TNFα results in EC apoptosis [72]
*MYLK* transcription in lung EC is increased by TNFα [61]	TNFα induces MMP-2, MMP-9 important in aneurysmal formation [42,73,84] MMP-2 and MMP9 are involved in BS (MMP-9 mainly in vascular BS) [74]
VEGF is involved in cell migration, proliferation, and angiogenesis and increases EC permeability [40]. VEGF increases the *MYLK* gene product and nmMLCK enzymatic activity involving the Sp1 transcription factor [40]	VEGF levels are increased in BS, mostly in vascular BS, correlated with disease activity and possibly predicting thrombosis [77,78]
ERK signaling is involved in the MLCK/MLCP balance by inhibiting MLCP [83]	ERK 1/2 in EC is increased in BS, stimulated by anti-endothelial antibodies [85] ERK activates MMPs [82]
*MYLK* is involved in inflammatory responses (apoptosis, vascular permeability, leukocyte diapedesis) [45] Activated neutrophils induce MLCK phosphorylation, and thus EC contractility and neutrophil migration [66,67] *MYLK* triggers neutrophil transmigration during acute lung injury by activating integrin-β2 [68]	Adherent neutrophils activate endothelial MLCK, and neutrophils are activated in BS [66]
MLCK pathway is involved in mediating proinflammatory cytokines (IL1β, IL-6, IL-8) expression [55]	Proinflammatory cytokines including IL-1β, IL-6, IL-8 are involved in BS pathogenesis [72]

Legend: AAA—abdominal aortic aneurysm, BS—Behçet syndrome, EC—endothelial cell, MMP—metalloproteinase, ERK1/2—extracellular signal-regulated kinase 1/2, HSS—Hughes-Stovin syndrome, PYK2—neutrophil tyrosine kinase, TNF-α: tumor necrosis factor-alpha, VEGF—vascular endothelial growth factor, sVEGFR—1 soluble VEGF—receptor 1.

As the *MYLK* mutation in our case interests the codon 491 involved in the EC cytoskeletal functions, it may take part in the processes discussed [23].

### 3.3. Other Possible Common Pathogenic Mechanisms in BS and Aneurysms

Many inflammatory cells in the aneurysmal tissue produce cytokines and enzymes promoting ECM degradation, depletion of SMCs, and vessel wall injury and remodeling [86]. A genome-wide association study in BS identified genes involved in focal adhesion, MAPK signaling, transforming growth factor beta (TGF-β) signaling, ECM-receptor interaction, and complement and coagulation cascades [80,87], suggests their involvement in pathogenesis (Table 2). For instance, shared genes between BS and aneurysms involving the ECM include the TGFβ/SMAD signaling pathway, active in BS [80,88,89], but also in TAAD and ICA [29,90]. Also, *ACTA 2*, involved in TAAD [55], is overrepresented in BS monocytes in the epithelial adherence junctions signaling [89].

The Notch pathway regulates developmental cell-fate decisions, modulates innate and adaptive immune responses, and is critical for vascular integrity maintenance and repair [91,92]. Notch1 haploinsufficiency causes TAAD in mice [93]. Notch1 is activated in active BS, likely related to decreased miR-23b expression [94]. Of interest, the decreased miR-23b also promotes aortic aneurysm formation by increasing the transcription of *FOXO4* (transcription factor forkhead box 4) involved in SMC phenotyping switching [95].

HLA-B51+, also present in our patient, confers an odds ratio of 5.9 to develop BS but accounts for only 20% of the genetic risk in BS [72,96]. Although the altered HLA-B51 peptide presentation is important in BS pathogenesis, the vasculitis seems not to be HLA-B51-related [22,96,97]. To our knowledge, HLA-B51 has not been often tested in HSS, as only one patient was tested in an earlier series, who was positive [1,98]. Nevertheless, HLA B51 may be present in HSS with or without other BS signs [99,100]. HLA-B51 includes a Bw4 epitope that interacts with the Killer cell immunoglobulin-like receptors KIR3DL1/DS1 on the NK cell surface [101]. *KIR3DL1/DS1* functional polymorphisms are found in BS [102]. KIR3DL1/DS1 and their HLA-class I ligands are associated with aneurysm formation in abdominal aortic aneurysms (AAA) [86].

Endoplasmic reticulum aminopeptidase 1 (ERAP1) may reflect common mechanisms with spondylarthritis by trimming the antigenic peptides to be loaded onto MHC class I molecules [21]. Although ERAP1 interacts with HLA-B51, it is not associated with vasculitis in BS [103,104].

Cytokines contribute to inflammation in BS and aneurysms genesis, respectively (Table 2) [72,80,90,105,106,107,108,109,110,111]. Also, the T helper cells Th1, Th2, and Th17 and their secreted cytokines are dysregulated in thoracic aortic aneurysms and dissections (Table 2) [59].

**Table 2 ijms-24-03160-t002:** Other possible common mechanisms involved in Behçet syndrome and aneurysms.

Mechanism/Pathway	Behçet Syndrome	Aneurysms
TGFβ	The TGF/SMAD3 pathway is overactive in BS [80,88] TGF-β1 increases in pulmonary vessels after mechanical stretching [75]	SMAD3 is involved in TAAD and Loeys-Dietz syndrome type III [29] TGF-β1 increases in ICA [90]
ECM-receptor interactions	COL1A2, COL5A1 are involved in BS [80]	COL1A2 and COL5A1 are involved in syndromic TAAD [29]
Proteasome	PSMA6 is found in GWAS in BS [80]	PSMA6 is involved in AAA [87]
Notch signaling	Notch 1 is involved in immune cells differentiation and activation [92] Notch1 is activated in BS [94]	Notch pathway is critical for integrity [91] Notch1 haploinsufficiency causes TAA in mice [93]
Mitogen-activated protein kinases (MAPK)	ERK 1/2 in EC is increased in BS, stimulated by anti-endothelial antibodies [81]	ERKs activate MMPs during AAA formation [82]
Interleukins	In BS TNFα, IL6, IL12/IL23 and IL10 are increased [80,106] IL-32 is involved in endothelial inflammation and coagulation in BS [109,110]	TNFα increases in ICA [90] IL6 increases in AAA [107] IL12/IL23 increases in AAA [108] IL10 increases in ICA [90] IL32 increases in AAA [111]
Regulation of IFNγ production and JAK/STAT signaling	IRF8 and IFNGR1 are involved in BS [106]	IFNγ is involved in ICA [90], and in experimental AAA [112], and JAK/STAT pathway in AAA [107]
VEGF	VEGF is increased in BS [78]	VEGF is increased in AAA [107]
MMP	MMP2 and MMP9 are involved in BS [74,75]	MMP2, MMP9 are increased in AAA [107]
Killer cell immunoglobulin-like receptors	KIR3DL1/DS1 polymorphisms are found in BS and interact with NK cells [100,101]	KIR3DL1/DS1 is associated with AAA formation [86]
Heat shock proteins	HSP60, HSP70 on Chlamydia, Mycoplasma involved in BS [72]	HSP60 and HSP70 bind to EC and macrophages in AAA [113]
Extracellular vesicles	EV are increased in BS predisposing to thrombosis [114]	EV mediates intercellular communication in aneurysm genesis [115]

Legend: AAA—abdominal aortic aneurysm, EV—extracellular vesicles, ICA—intracerebral aneurysm HSP—heat shock proteins, IFNGR1—Interferon Gamma Receptor 1, IFNγ-interferon gamma, KIR3DL1-Killer cell immunoglobulin-like receptors, MMP—metalloproteinase, PSMA6—Proteasome 20S Subunit Alpha 6, TAAD—thorarcic aortic aneurysm and dissection, VEGF—vascular endothelial growth factor. Note: Very few patients studied had vascular BS; for other genes apart from HLA-B regions, the effect sizes are small, and the functional consequences of most genetic variations in BS pathogenesis are unknown.

Molecular mimicry for antigens such as *Chlamydia pn*., *Mycoplasma* spp. *S. sanguis*, *H. pylori*, *Staph aureus*, bearing autoantigens such as the heat shock proteins HSP60, HSP70, was found in BS [72]. *Chlamydia* and *Mycoplasma* initially colonize the adventitia through vasa vasorum [116], whereas HSP60 and HSP 70 bind to EC and macrophages and induce the secretion of proinflammatory cytokines and MMPsin AAA [113]. Extracellular vesicles (EV), membrane-surrounded particles, modulate inflammation, vascular dysfunction, and thrombosis [117]. EV are involved in aneurysm pathogenesis, mediating intercellular communication [115]. Moreover, EV are increased in BS, predisposing to thrombosis [114].

From the actors participating in the complex shared mechanisms of BS and aneurysms, our patient was HLA-B51-positive.

### 3.4. Other Potential Contributors to Aneurysm Formation in Our Case

Several other genetic factors raised questions regarding the potential contribution to the occurrence of PAA.

The heterozygous low-penetrance *CTFR* pathogenic mutation may be associated with cystic fibrosis, with congenital bilateral absence of the vas deferens, and in heterozygous carriers with increased risk for pancreatitis. Cystic fibrosis, an autosomal recessive (AR) disease beyond the status of the carrier, may result in haploinsufficiency, increasing the risk for cystic fibrosis-related conditions [118]. Noteworthy, PAA and bronchial artery aneurysms have been described in cystic fibrosis [119]. The *CTFR* carriage would also increase the risk of pancreatitis and gastrointestinal cancers, which is important in long-term management under azathioprine [118].

The patient had a heterozygous pathogenic mutation in *CR2*, encoding the complement C3d receptor 2, a membrane protein functioning as a receptor for the Epstein-Barr virus on B and T lymphocytes, which also inhibits IL-6 production [120]. *CR2* mutations may be associated with a type of AR common variable immunodeficiency and autoimmune diseases due to the impairment of self-tolerance [120]. Both cystic fibrosis and common variable immune deficiency may be associated with bronchiectasia, a cause of hemoptysis [121].

Mutations of *GGCX* encoding an enzyme involved in the metabolism of Gla proteins may also cause an AR pseudoxanthoma elasticum-like disorder with multiple coagulation factors deficiency, and at times with vascular abnormalities, including cerebral aneurysms or pulmonary artery stenosis [122,123]. Haploinsufficiency has been described for *GGCX* carriers as well [123].

Although a single copy of *NPH2* is unlikely to create the AR dyskeratosis congenita, and the patient has no clinical features to support the diagnosis of this telomere disorder, dyskeratosis congenita may be associated with pulmonary arterio-venous malformations and with bone marrow failure [124]. *FANCE* may be associated with AR Fanconi’s anemia, also a cause of arteriovenous pulmonary fistulae [125].

BS is a multifactorial polygenic disease, with genetic, epigenetic, environmental, and immunological contributors [126]. Inflammation plays a major role in BS pathogenesis [127]. The genetics of BS are complex, involving more than one pathogenic pathway [126,127]. Nevertheless, BS in the same family seems not to accumulate in similar clinical clusters [128]. Moreover, different vessels may be involved in BS relapses [20]. This would plead for the outstanding role of multiple non-genetic factors in BS relapses [128]. However, in an individual BS patient, the genetic background may contribute to the shaping of the clinical disease appearance.

### 3.5. Therapy

In our patient, the left pulmonary artery aneurysm decreased in size, likely because of the cortisone treatment for COVID infection.

Vascular BD or HSS respond generally to glucocorticoids and cyclophosphamide, or anti-TNFα in refractory cases, or in cases with pulmonary vessel involvement [19,71,129]. In BS, aneurysms may develop at the site of arterial puncture. Surgical PAA repair carries a high risk of massive hemoptysis, and arterial embolization with catheter angiography may be an emergency alternative [130]. In BS, except for in venous cerebral thrombosis therapy, anticoagulation is less effective than immunosuppression in preventing recurrent thrombosis [71]. Anticoagulation may be necessary, often in the coexistence of cardiac thrombus, but is risky in the context of PAA and should parallel immunosuppression [22,131].

Of interest, MLCK is a potential therapeutic target for inflammatory diseases [132]. The VEGF-induced nmMLCK expression and EC permeability can be attenuated by silencing the transcription factor Sp1 [88]. Nevertheless, inhibiting targets such as VEGF or Notch should be weighed against the possible deleterious effects [92].

## 4. Conclusions

The different BS phenotypes are likely based on different genetic determinants. As such, HSS may be a vascular BS in the presence of sometimes minor gene variants resulting in disruption of vascular organization, SMC loss, contractile dysfunction, and formation of aneurysms [50]. Several other gene variants involved in angiogenesis, arterial dissections, or thrombosis may contribute to shaping the vascular BS phenotype. Nevertheless, BS is a polygenic disease with genetic, epigenetic, environmental, and immunological contributors, and some findings from TAAD cannot be simply extrapolated [126]. However, deciphering the specific pathogenic contributors in an individual BS patient may help improve disease understanding [133].

Managing BS and its specific variants is complex and challenging [134]. In patients with BS, a hemorrhage should inspire a suspicion of HSS [129]. Improvement of diagnostic techniques may aid in reaching a rapid diagnosis which may be life-saving in this setting [135,136]. Noteworthy, the PA wall thickness is increased in BS with major organ involvement, which could be important also for diagnosis in cases with incomplete presentation [137]. Besides, new findings regarding aneurysm formation could advance pharmacological interventions [133]. Recent advances in diagnostic techniques allow an early diagnosis of a specific Behçet syndrome subtype and other associated conditions to personalize the disease management. In the presented HSS case, actually a vascular BS, several variants of genes involved in angiogenesis were found. Genetic testing in other HSS cases could help identify the mechanisms underlying the PAA formation besides pulmonary vasculitis.

## Figures and Tables

**Figure 1 ijms-24-03160-f001:**
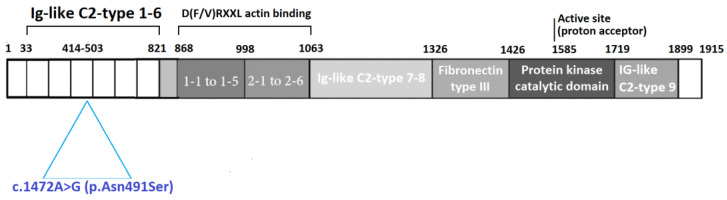
A graphic illustration of the full-length human *MYLK* gene, structural model consisting of 1915 amino acids with domains; this gene, a muscle member of the immunoglobulin gene superfamily, encodes myosin light chain kinase (a calcium-calmodulin dependent enzyme). Also regulates actin-myosin interaction through a non-kinase activity. Depicted are the actin-binding domain, catalytic core, the regulatory segment containing the inhibitory and calmodulin-binding domains, and the kinase-related protein (KRP) domain. The region where the patient’s variant c.1472A>G (p.Asn491Ser) is located is indicated by the blue triangle. In our case, the *MYLK* mutation interests the codon 491, localized in the exon 11, in the Ig-like domain 3, involved in the EC cytoskeletal functions based on [23].

**Figure 2 ijms-24-03160-f002:**
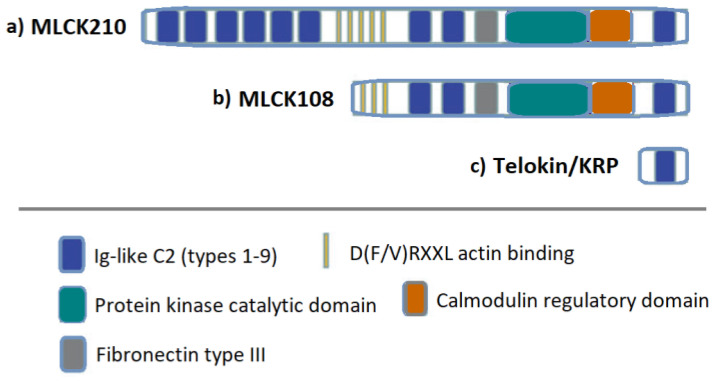
Myosin Light Chain Kinase gene (*MYLK*) products; schematic representation of each protein and its domain structure. The gene *MYLK* encodes 3 proteins: MLCK210 (210–220 kDa), MLCK108 (110–140 kDa), and telokin/kinase-related protein (KRP) (based on [44,46]).

**Figure 3 ijms-24-03160-f003:**
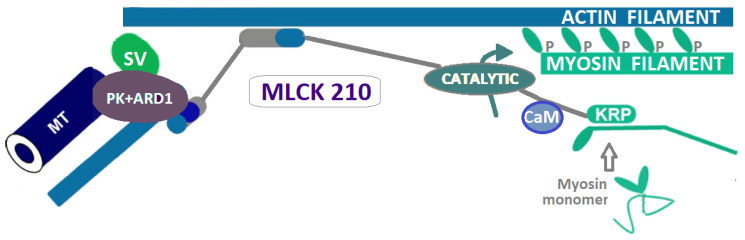
MLCK210 is a signal integrator molecule containing several interaction sites for cytoskeletal and regulatory proteins (based on [44,45]). Adapted with permission from Ref. [44], Shirinsky, V.P. (2012). MYLK (Myosin Light Chain Kinase). In: Choi, S. (eds) Encyclopedia of Signaling Molecules. Springer, New York. https://link.springer.com/referenceworkentry/-10.1007/978-1-4419-0461-4_248#citeas, license number 5481371185567/2023, Legend: CaM—calmodulin, KRP—kinase-related protein domain, MT—microtubules, PK+ARD1—Protein kinases and ARD1 acetylase that modify MLCK210 residues, SV—supervillin (a membrane-associated scaffolding protein interacting with MLCK210 N-terminus and with myosin II).

## Data Availability

Documentary evidence regarding the patient’s data is not publicly available for confidentiality reasons.

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
