# Peer review of "Frail Silk: Is the Hughes-Stovin Syndrome a Behçet Syndrome Subtype with Aneurysm-Involved Gene Variants?"

_ijms, 2023, doi:10.3390/ijms24043160_

Round 1

Reviewer 1 Report

The present work is aimed at discussing the possible involvement of the genetic findings, as well as other potential common determinants of Behçet/Hughes-Stovin syndrome and aneurysms in vascular Behçet syndrome. The case and the review is well discussed. I only would suggest adding few main messages for the reader at the end of the conclusions.

Take in to consideration the potential role of genetic in the Hughes-Stovin syndrome secondary to Behçet syndrome subtype 
with aneurysm.

It is important to improve the knowledge of such a rare condition as subtype of a rare disease.

A well reported discussion and analysis of the case.

Please include any additional comments on the tables and figures.

Author Response

The present work is aimed at discussing the possible involvement of the genetic findings, as well as other potential common determinants of Behçet/Hughes-Stovin syndrome and aneurysms in vascular Behçet syndrome. The case and the review is well discussed.

We thank the Reviewer for the kind comments.

I only would suggest adding few main messages for the reader at the end of the conclusions.

We thank the Reviewer for the kind suggestion. We have added the main messages at the end of the conclusions as suggested.

Take into consideration the potential role of genetic in the Hughes-Stovin syndrome secondary to Behçet syndrome subtype with aneurysm. It is important to improve the knowledge of such a rare condition as subtype of a rare disease.

We truly thank the Reviewer for the considerate comment.

A well reported discussion and analysis of the case.

We thank the Reviewer for the kind words.

Please include any additional comments on the tables and figures.

We thank the Reviewer for the pertinent suggestion. We have completed the Legends of the tables and clarified the legends of the figures as suggested.

Reviewer 2 Report

Manuscript ID ijms-2131147"Frail Silk: Is the Hughes-Stovin syndrome a Behçet syndrome subtype with aneurysm-involved gene variants?”

A very interesting and beautiful article.

The presented patient is actually a complete Behçet's Disease patient.

However, why do some patients with Behçet's disease developing aneurysm, while others do not? I think it is an important article in terms of seeking an answer to this question. A very comprehensive analysis has been made. I will not make any additional suggestions.

Author Response

A very interesting and beautiful article.

We heartly thank the Reviewer for the kind words.

The presented patient is actually a complete Behçet's Disease patient.

However, why do some patients with Behçet's disease developing aneurysm, while others do not? I think it is an important article in terms of seeking an answer to this question.

We thank the Reviewer for the considerate comments.

A very comprehensive analysis has been made. I will not make any additional suggestions.

We are grateful to the Reviewer for the appreciation.

Reviewer 3 Report

Frail silk: is the Hughes-Stovin syndrome a Behcet syndrome subtype with aneurysm-involved gene variants:

Thank you for the opportunity to review this manuscript describing a unique case presentation and literature review of Hughes-Stovin syndrome, particularly focusing on genetic variations that may contribute. Comments mainly focus on restructuring throughout to improve clarity and enhance impact of information presented. 

35 – define Behcet syndrome (BS) here – is used as an acronym in line 36 but not defined previously

37, 41 – continue to use BS as acronym if defined for consistency

57 – would perhaps be good to specify here that you are referring to HSS (not BS?)

68 – aneurysm?

68 – two instances where tracked changes are still present

68 – comma after Nevertheless, 

71 – comma after BS,

75 – artery not arteries?

77 – another work besides interest is probably more appropriate – involve?

78 – In HSS, histology reveals…

83 – tracked changes present

87 – large vessel involvement predominant in the clinical picture

Overall, the introduction contains some important information but would perhaps be more clear with some restructuring: recommend to move paragraph 3 on etiology/pathogenesis after paragraph 1 and combine into one paragraph. Then combine current paragraph 2 and 4 on diagnostic criteria and prevalence of findings consistent with disease diagnosis into the second paragraph overall. Then combine paragraph 5 and 6 into one paragraph on histologic characteristics. Finally end with paragraph 7 but expand upon it to guide the reader into the remainder of the article: Therefore, the objective of this case report was… Here we report... key insights.

98 – perhaps this would be obvious to other readers, but unclear to me what 7-pack years indicates? Could you please make this more apparent?

107 – with an adherent thrombus? Or thrombi? Singular or pleural?

113, 120, 121, 133,134, 140 – where drugs are named, it would be useful to the reader to list drug class and guide the reader with some rationale for implementation at that stage

129-132 – the average reader may not understand the relevance of these statements – guide the reader through with statements, such ‘… indicating XYZ…’

Figure 1 legend – from not form?

Figure 2, 3 – add a and b to figures themselves (assuming a is left and b is right but this is not listed)

Discussion does become somewhat acronym heavy which is understandable – this could perhaps be handled using a table defining acronyms etc for the reader to refer back to

217 – aneurysm formation involves

232 and 236 – combine these paragraphs? 

Overall, the discussion reads more as a literature search with descriptive information rather than a discussion of what is known about the disease process and genetic background related to the case report provided. As a result, while there is interesting and important information included, this section comes across as somewhat disjointed and it is difficult to follow as well as interpret the significance of individual pieces of information provided. Conclude each section with a summary statement and link it back to the case presented. Finally, the discussion switches between first and third person at times; recommend to maintain consistency throughout.

Conclusions – recommend to restructure this – move paragraph 2 after paragraph 3 and combine into one paragraph with paragraph 4 making the conclusions section 2 paragraphs overall. Finally, end with a stronger statement of summary as the section somewhat trails off with “Also…’

Author Response

Thank you for the opportunity to review this manuscript describing a unique case presentation and literature review of Hughes-Stovin syndrome, particularly focusing on genetic variations that may contribute.

 We thank the Reviewer for the kind comments.

Comments mainly focus on restructuring throughout to improve clarity and enhance impact of information presented. 

35 – define Behcet syndrome (BS) here – is used as an acronym in line 36 but not defined previously

We thank the Reviewer for the kind suggestion and for helping us improve the consistency (indeed we have inappropriately used an abbreviation in the abstract; we have changed it at line 36 and then in the main text).

37, 41 – continue to use BS as acronym if defined for consistency

57 – would perhaps be good to specify here that you are referring to HSS (not BS?)

68 – aneurysm?

68 – two instances where tracked changes are still present

68 – comma after Nevertheless, 

71 – comma after BS,

75 – artery not arteries?

77 – another work besides interest is probably more appropriate – involve?

78 – In HSS, histology reveals…

83 – tracked changes present

87 – large vessel involvement predominant in the clinical picture

 We thank the Reviewer for the pertinent observations. We have changed the text accordingly.

Overall, the introduction contains some important information but would perhaps be more clear with some restructuring: recommend to move paragraph 3 on etiology/pathogenesis after paragraph 1 and combine into one paragraph. Then combine current paragraph 2 and 4 on diagnostic criteria and prevalence of findings consistent with disease diagnosis into the second paragraph overall. Then combine paragraph 5 and 6 into one paragraph on histologic characteristics. Finally end with paragraph 7 but expand upon it to guide the reader into the remainder of the article: Therefore, the objective of this case report was… Here we report... key insights.

We thank the Reviewer for the kind suggestions. We have changed the text accordingly.

98 – perhaps this would be obvious to other readers, but unclear to me what 7-pack years indicates? Could you please make this more apparent?

We thank the Reviewer for the comment. We have clarified the text regarding the pack years.

107 – with an adherent thrombus? Or thrombi? Singular or pleural?

We thank the Reviewer for the observation. We have corrected the text regarding the thrombi.

113, 120, 121, 133,134, 140 – where drugs are named, it would be useful to the reader to list drug class and guide the reader with some rationale for implementation at that stage.

We thank the Reviewer for the suggestion. We have craified the text accordingly.

129-132 – the average reader may not understand the relevance of these statements – guide the reader through with statements, such ‘… indicating XYZ…’

We thank the Reviewer for the comment. We have clarified the text as indicated.

Figure 1 legend – from not form?

We thank the Reviewer for the observation. We have corrected the text.

Figure 2, 3 – add a and b to figures themselves (assuming a is left and b is right but this is not listed)

Please kindly note that the figures are not anymore part of the manuscript, upon the Editor’s requirement.

Discussion does become somewhat acronym heavy which is understandable – this could perhaps be handled using a table defining acronyms etc for the reader to refer back to.

We thank the Reviewer for the pertinent comment. We have included an acronym list at the end of the manuscript.

217 – aneurysm formation involves

232 and 236 – combine these paragraphs? 

 We thank the Reviewer for the insightful comments. We have corrected the text as suggested.  

Overall, the discussion reads more as a literature search with descriptive information rather than a discussion of what is known about the disease process and genetic background related to the case report provided. As a result, while there is interesting and important information included, this section comes across as somewhat disjointed and it is difficult to follow as well as interpret the significance of individual pieces of information provided. Conclude each section with a summary statement and link it back to the case presented.

We thank the Reviewer for the insightful comments. We have done as kindly suggested.

Finally, the discussion switches between first and third person at times; recommend to maintain consistency throughout.

We thank the Reviewer for the pertinent observation. We have changed the text accordingly.

Conclusions – recommend to restructure this – move paragraph 2 after paragraph 3 and combine into one paragraph with paragraph 4 making the conclusions section 2 paragraphs overall. Finally, end with a stronger statement of summary as the section somewhat trails off with “Also…’

We thank the Reviewer for the thoughtful comment. We have changed the text as suggested.

We have included a table with abbreviations in the end of the manuscript.

Reviewer 4 Report

The main proposal of the study did not have enough rationale and background supporting data. The conclusion is also not proper in my opinion. The evidence provided is not compelling. Please see the attachment for detailed comments.

Author Response

The main proposal of the study did not have enough rationale and background supporting data. The conclusion is also not proper in my opinion. The evidence provided is not compelling. Please see the attachment for detailed comments.

We thank the Reviewer for the comments and for kindly having thoroughly reviewed our manuscript.

Regarding these commentaries on the text, please kindly see below our answers:

  1. PAA in HSS usually involve the interlobar, lobar and segmental branches. Main PA involvement is present only in a minority of cases. This statement should be omitted to avoid confusion.

We thank the Reviewer for the pertinent comment. We have omitted the specific sentence as suggested.

  1. It would be nice to say glucocorticoids and cyclophosphamide since cyclophosphamide is almost never used without glucocorticoids.

We thank the Reviewer for the good observation. We have changed the text accordingly (mostly in the Section 3.5 regarding the therapy).

  1. It is not necessary due to bronchial artery involvement in angiodysplasia. Lobar/segmental PA vasculitis without angiodysplasia may cause both thrombosis and hemorrhage in HSS.

We thank the Reviewer for the pertinent comment. We have omitted the sentence as suggested.

  1. Although it may be hypothesized as such, no vascular dysplasia genetic risk factors were identified in Behcet’s syndrome to date. Spondylarthritis susceptibility genes were also not studied in articular phenotype of BS patients but in BS in general. So, it is actually not known.

We thank the Reviewer for the comment. We have completed the sentence in the text” “although no such genetic risk factors were identified in BS to date”.

  1. It is a quite risky approach for fatal hemoptysis. Vascular BS patients, as in this case, do not benefit from anticoagulation. Hemoptysis episode in this case may actually be caused by anticoagulant and antiplatelet agents.

We thank the Reviewer for the insightful comment. We agree, as stated in the manuscript that in vascular BS patients the thrombosis is secondary to inflammation, and the anticoagulation may be  necessary, often in a context of a cardiac thrombus, but is risky in the context of pulmonary artery aneurysms (please see Section 3.5).   However, in this patient’s case the decision to cautiously anticoagulate was an informed shared decision of his cardiology medical team, which considered that in the peculiar patient’s case the benefits of anticoagulation outweigh the risks. After the diagnosis and introduction of  immunosuppression anticoagulation was reduced to a minimum, but still kept as the patient had an inferior vena cava thrombosis extended to the left renal vein, persisting despite immunosuppression.  

  1. But could not be shown to date as stated previously.

We thank the Reviewer for the comment. We have completed the text with this observation.

  1. But PA is more likely a vein rather than an artery/ It does not have a thick muscular wall and a high pressure inside. Vascular BS patients with severe pulmonary involvement usually have concurrent or past large vein involvement. In those without large vein involvement, popliteal and iliac vein wall thickness was shown to be increased.

We thank the Reviewer for the astute observation. We have completed the text regarding the examination.

  1. TNF-alpha is a common endothelial cell activator resulting in activator-induced apoptosis in endothelial cells. This is not a BS-related feature.

We thank the Reviewer for the rightful common. We have omitted ” in BS” from the sentence.

  1. Very few patients in these studies had vascular BS. Additionally, although of statistical significance, the effect sizes (i.e.the odds ratios) of having genetic perturbations other than the HLA-B regions are quite small. The functional consequences of most of these genetic variations in the context of BS pathogenesis are also not known.

We thank the Reviewer for the significant observation, which we have included in the manuscript.

  1. This statement cannot be a conclusion of the review.

We thank the Reviewer for the pertinent observation. We have modified the conclusions accordingly.

Reviewer 5 Report

I would like to congratulate authors for their excellent case presentation of the patient with Hughes-Stovin syndrome. Authors provide data retrieved not only from instrumental examination but from genetic tests as well. Despite this interesting and complete presentation of such rare case I have some comments down below:

Abbreviation BS should be deciphered

Line 55: brackets should be round

Please, revise the main text. Some abbreviation explanations are repeated (for example, PAA, ICA, VUS etc.), some explanations are not necessary because of  single occurrence (for example, LLL).

What 2019 mean in the description of the Figure 1?  Year?  Then I strongly recommend to use the timeline in the text and figures. Do not incorporate date of the examination in the figure. Place it in the figure description section.

Please use unified terminology of the examination tests throughout the manuscript. For example, Thoracic angio CT or angio CT or thoracic CT angiogram

In the discussion (not discussions) section I would recommend to remove 3rd and 4th paragraphs. Information is irrelevant to the article. Moreover, this unjustifiably enlarge the manuscript. If authors wanted to emphasize genetic role in HSS they should shortly describe including related data only.

What is the link between figure 1 and the role of the SMC in the discussion section? Please, correct this discrepancy

What is the need to describe physiological mechanisms occurring in ECM in the context of the article?

Line 281: What is the TADD?

To my mind, the therapy section is out of the point. Possibly this part of the text is unnecessary.  

The text is overloaded by redundant information (e.g. function of each enzyme, its taxonomy) and, consequently, by abbreviations. It is quite hard to follow authors thoughts. Please, remove the irrelevant data.

In my opinion, authors should not include all the recent information about BS and HSS. I think it will be way better if authors will discuss existing data relevant to their case presentation. And this is partly demonstrated in the 3.4. Other potential contributors to aneurysm formation in our case section .

Author Response

I would like to congratulate authors for their excellent case presentation of the patient with Hughes-Stovin syndrome. Authors provide data retrieved not only from instrumental examination but from genetic tests as well.

We heartly thank the Reviewer for the kind comments.

Despite this interesting and complete presentation of such rare case I have some comments down below:

 Abbreviation BS should be deciphered.

 Line 55: brackets should be round.

We thank the Reviewer for the observations. We have corrected the text accordingly.

 Please, revise the main text. Some abbreviation explanations are repeated (for example, PAA, ICA, VUS etc.), some explanations are not necessary because of single occurrence (for example, LLL).

 We thank the Reviewer for the astute observation. We have revised the text and corrected accordingly.

What 2019 mean in the description of the Figure 1?  Year?  Then I strongly recommend to use the timeline in the text and figures. Do not incorporate date of the examination in the figure. Place it in the figure description section.

 We thank the Reviewer for the very good observation. Please kindly note that the former Figure 1 (removed upon submission at the Editors’ request according to the Journal’s requirements) is no longer present in the text and faulty remained in the manuscript variant submitted for review.

Please use unified terminology of the examination tests throughout the manuscript. For example, Thoracic angio CT or angio CT or thoracic CT angiogram.

  We thank the Reviewer for the insightful comment. We have unified the terminology as suggested.

In the discussion (not discussions) section I would recommend to remove 3rd and 4th paragraphs. Information is irrelevant to the article. Moreover, this unjustifiably enlarge the manuscript. If authors wanted to emphasize genetic role in HSS they should shortly describe including related data only.

We thank the Reviewer for the pertinent observation. We have consistently shortened the paragraph 4.

What is the link between figure 1 and the role of the SMC in the discussion section? Please, correct this discrepancy.  

We thank the Reviewer for the observation. The Figure 1 in the original submission (consisting in patient’s imaging) was removed at the Editors’ suggestion, as per the Journal’s requirements. We will remove all the referrals to this figure.

What is the need to describe physiological mechanisms occurring in ECM in the context of the article?

 We thank the Reviewer for the insightful observation. We only wanted to stress the involvement of MYLK in the extracellular matrix as well.

Line 281: What is the TADD?

We thank the Reviewer for the observation. We have corrected TADD replacing it with TAAD (thoracic aortic aneurysm and dissection).

To my mind, the therapy section is out of the point. Possibly this part of the text is unnecessary.

We thank the Reviewer for the pertinent comment. We have copiously shortened the therapy section.

The text is overloaded by redundant information (e.g. function of each enzyme, its taxonomy) and, consequently, by abbreviations. It is quite hard to follow authors thoughts. Please, remove the irrelevant data.

We thank the Reviewer for the observation. We have removed the redundant data accordingly.

 In my opinion, authors should not include all the recent information about BS and HSS. I think it will be way better if authors will discuss existing data relevant to their case presentation. And this is partly demonstrated in the 3.4. Other potential contributors to aneurysm formation in our case section

We thank the Reviewer for the pertinent comment. We have simplified the discussion as suggested.

We are deeply grateful to the Reviewers for their time and for the very insightful comments, which helped us to improve our work.

Round 2

Reviewer 3 Report

The authors appear to have addressed this reviewer's comments and the manuscript is improved in its current form.

Reviewer 4 Report

The manuscript improved after signficant effort of the authors. I have no further comments.